# Coming in the Air: Hypoxia Meets Epigenetics in Pancreatic Cancer

**DOI:** 10.3390/cells9112353

**Published:** 2020-10-25

**Authors:** Claudia Geismann, Alexander Arlt

**Affiliations:** 1Laboratory of Molecular Gastroenterology & Hepatology, Department of Internal Medicine I, UKSH-Campus Kiel, 24105 Kiel, Germany; cgeismann@email.uni-kiel.de; 2Department for Gastroenterology, European Medical School (EMS), Klinikum Oldenburg AöR, 26133 Oldenburg, Germany

**Keywords:** pancreatic cancer, hypoxia, HIF, epigenetics, DNA methylation, non-coding RNA, histone modifications

## Abstract

With a five-year survival rate under 9%, pancreatic ductal adenocarcinoma (PDAC) represents one of the deadliest tumors. Although the treatment options are slightly improving, PDAC is the second leading cause of cancer related death in 2020 in the US. In addition to a pronounced desmoplastic stroma reaction, pancreatic cancer is characterized by one of the lowest levels of oxygen availability within the tumor mass and these hypoxic conditions are known to contribute to tumor development and progression. In this context, the major hypoxia associated transcription factor family, HIF, regulates hundreds of genes involved in angiogenesis, metabolism, migration, invasion, immune escape and therapy resistance. Current research implications show, that hypoxia also modulates diverse areas of epigenetic mechanisms like non-coding RNAs, histone modifications or DNA methylation, which cooperate with the hypoxia-induced transcription factors as well as directly regulate the hypoxic response pathways. In this review, we will focus on hypoxia-mediated epigenetic alterations and their impact on pancreatic cancer.

## 1. Introduction

In western countries cancer-related diseases are a major cause of mortality. Pancreatic ductal adenocarcinoma (PDAC) represents one of the deadliest tumor diseases with a five-year survival rate of 2–9% and, unlike other cancer entities, PDAC exhibits rising incidence rates [1]. Projections are assuming that until 2030 PDAC becomes the second leading cause of cancer-related death in the US and Western Europe [2,3]. Although new therapy options are investigated, among them the introduction of new surgical techniques and improvements in first and second-line palliative therapies, poor response-rates and the acquirement of chemoresistance represent serious obstacles in the treatment of PDAC patients.

PDAC is characterized by an intense desmoplastic stroma reaction as well as a severe hypo-vascular environment, contributing to the development of a tumor-mass with a low availability in nutrients and oxygen. In comparison to other solid tumors, PDAC show one of the lowest levels of oxygen availability [4,5]. In this context, direct measurements of oxygenation in human PDAC tumor tissues displayed a profound reduced partial pressure (pO_2_) compared to the adjacent normal tissue (median pO_2_ pressure of tumor mass: 0–5.3 mmHg; median pO_2_ of normal adjacent tissue: 9.3–92.7 mmHg) [6,7,8]. This condition of a deprived adequate oxygen supply at tissue level is denoted as hypoxia. Hypoxia affects almost all hallmarks of cancer and contributes to a multitude of cellular functions mediating therapy resistance, aggressiveness and metastasis of tumor cells [9,10]. Increasing evidence suggests that the hypoxic response is codetermined by epigenetic changes in solid tumors, setting the focus of this review on epigenetic changes in PDAC due to hypoxic conditions.

For this review, we searched PubMed and Google Scholar with the keywords PDAC, epigenetics and hypoxia until September 2020 and discussed the received citations carefully.

## 2. Hypoxia in Cancer

In order to enable continuous rapid cancer cell proliferation and to overcome a lack of nutrient and oxygen availability within the tumor mass, cellular signaling pathways are altered in cancer cells compared to non-transformed cells. In this context, the hypoxic cellular response is primarily regulated through the hypoxia-inducible transcription factor family (HIF) which mediates gene regulation affecting angiogenesis, metabolism, migration, invasion, immune escape and resistance to therapy.

To counteract an undersupply of oxygen and the related limitations for further cell proliferation, a vascularization of the tumor mass is triggered by the secretion of various pro-angiogenic factors including vascular endothelial growth factor (VEGF), matrix metalloproteinase-9 (MMP-9)**,** interleukin-8 and fibroblast growth factor-2 [11]. While in normal tissues a highly regulated interplay of pro and anti-angiogenic factors determines the formation of functional vessels, tumor associated vessels exhibit often a chaotic architecture. In line, the oxygen supply through this tumor associated neoangiogenesis is often ineffective leading to tumor regions with moderate to severe hypoxia [12,13,14].

Next to an enhanced angiogenesis, tumor cells adapt to reduced oxygen availability by reprogramming metabolic pathways in order to minimize oxygen consumption and to ensure survival and proliferation of the cancer cells [15]. Hypoxic cancer cells show an enhanced uptake of glucose, elevated glycolysis and lactate production while simultaneously oxidation via tricarboxylic acid cycle and oxidative phosphorylation is reduced [16,17].

Furthermore, the hypoxic environment is beneficial for migration and invasion by facilitating the epithelial-mesenchymal transition (EMT) program in PDAC cells [18] by a differential regulation of the EMT-associated transcription factors Slug and Snail as wells as cell adhesion molecules [4,19].

Moreover, an altered function of the innate and adaptive immune cells is also part of hypoxia-mediated signaling modifications of both tumor cells and the stromal compartment. One option for a hypoxia-mediated immune escape is mediated by VEGF. As aforementioned, lowered oxygenation induces the translation and secretion of VEGF protein in both cancer cells and cells of the surrounding microenvironment. This secretion of VEGF as well as the upregulation of certain HIFs promote an attraction and differentiation of myeloid-derived suppressor cells (MDSC) to the tumor mass. MDSCs are known to facilitate tumor progression due to hypoxia by various immunosuppressive mechanisms, e.g., by the suppression of an anti-cancer T-cell response or the differentiation of MDSCs to M2-polarized macrophages [20,21].

Apoptosis resistant mechanisms of cancer cells are also affected by hypoxia [7] and a reduced oxygen availability affects chemoresistance in various ways: (1) transcription of multidrug resistance pumps is upregulated leading to a decreased intracellular chemotherapeutic drug concentration (2) cytotoxicity of a number of chemotherapeutics is affected by lowered oxygenation [4,8,22]. Furthermore, hypoxia promotes cell survival by inducing a quiescent state, inhibiting apoptosis (enhanced anti-apoptotic protein expression like B cell lymphoma-extra large (Bcl-xL), cellular inhibitor of apoptosis protein (cIAP), cellular FLICE inhibitory protein (c-FLIP)), controlling autophagy and influencing the immune response against cancer cells [8,9]. Likewise, the effects of radiotherapeutic treatment are highly affected under hypoxic conditions by a reduced DNA free radical availability and an increase in DNA repair enzyme [7,8].

## 3. HIF Transcription Factor Family—The Key Players in Response to Hypoxia

In all of the above described adaptions of cancer and environmental cells to the altered, hypoxic parameters the hypoxia-inducible factor transcription factor family plays a central role [23]. HIF is a heterodimer consisting of a ubiquitously expressed β-subunit (HIF-1β) and an oxygen sensitive α-subunit. The latter consist of three subunits HIF-1α, HIF-2α and HIF-3α. The function of HIF-3α is discussed contradictory, while for a long time HIF-3α was attributed to compete with HIF-1α and HIF-2α for binding to promoter regions and having a negative impact on hypoxia-inducible gene expression, recent published data showed a positive, hypoxia-dependent transcriptional regulation of HIF-3α target genes [24,25]. Zhou et al. were able to show a hypoxia-mediated expression of HIF-3α in pancreatic cancer tissues associated with increased distant metastasis and local invasion as well as reduced survival times [26]. HIF-3α possesses only one transactivation domain (TAD) and one prolyl site, while HIF-1α and HIF-2α share high homology and feature two prolyl sites and two TADs [26]. Similarities in the DNA-binding domains and heterodimerization domains of HIF-1α and HIF-2α suggest a unique target gene profile, although HIF-1α seems to mediate the acute response to hypoxia while HIF-2α stabilizes over longer time frames and under normoxic conditions [9,14]. 

In the regulation of HIF, a distinction between normoxic and hypoxic conditions can be made. Under normoxic conditions there are two crucial proline residues within the oxygen-dependent degradation domain of the HIF-α subunits hydroxylated by prolyl hydroxylase enzymes (PHD1, PHD2, PHD3) enabling an interaction with the Von Hippel-Lindau tumor suppressor protein (pVHL) and a subsequent recruitment of a E-3 ubiquitin ligase complex that conducts HIF-α to a ubiquitin-mediated proteasomal degradation [14]. In the absence of oxygen, the activity of PHDs is inhibited and leads to a stabilization and translocation of HIF-α into the nucleus. Nuclear HIF-α heterodimerizes with HIF-1β to form an active HIF enzyme, binds to hypoxia response elements (HRE) within HIF target gene promoters and induces subsequent the transcription of hypoxia responsive genes [8,9]. The activity of HIF-1α under hypoxic conditions is modulated by coactivation factors, e.g., the histone acetyltransferase enzyme CAMP response element binding protein (CREB) binding protein/p300 (CBP/p300). This interaction of HIF-1α with CBP/p300 can be impaired by factor inhibiting HIF (FIH) hydroxylation or the binding of pVHL protein that recruits histone deacetylase enzymes (HDAC) and inhibits the HIF-1α transactivation [27]. The direct interaction of epigenetic modifying enzymes with HIF-1α already points to a connection of hypoxic cell response and epigenetics and was strategic starting point for the following review about epigenetic regulation in dependence of hypoxia in pancreatic cancer.

## 4. Hypoxia Influenced Epigenetic Regulation of Pancreatic Cancer

While diverse studies in various cancer cells have shown an impact of hypoxia-mediated epigenetic mechanisms in the establishment of cancer hallmarks (comprehensively reviewed by Camuzi et al. [28]), the scientific exploration of epigenetic changes under hypoxic conditions in PDAC is still at the very beginning. These epigenetic changes are characterized on RNA basis, by the regulation of non-coding RNAs (microRNAs and long non-coding RNAs), on DNA basis, by (hydroxy)methylation of DNA as well as on protein basis, by posttranslational modifications of histones or the (in)activation of epigenetic regulator-proteins [29].

### 4.1. Hypoxic Regulation of miRNAs in PDAC Cells

MircoRNAs (miRNAs) are small, 18–24 nucleotide non-coding RNAs that predominantly regulate target gene expression by inhibiting RNA-translation through binding of their “seed region” to the complementary sequence within the 3’UTR of the target gene and/or induce their degradation [30,31,32]. The underlying mechanism of miRNA deregulation in cancer cells is at present only poorly understood and is very likely resulting from multiple mechanisms which can act individually but also collectively [33,34]. In this context, a regulation of miRNAs on transcriptional level is a possible explanation but also a deregulated biogenesis or maturation of miRNAs as well as regulation by epigenetic mechanisms have been detected and are potential mechanisms in tumor cells [34]. 

Bhandari et al. showed that the expression of 658 out of 784 (84%) measured miRNAs correlated with hypoxia in at least one tumor type and that the hypoxic impact on miRNA directionality was frequently consistent across tumor types. In particular, elevated miRNA-210 expression correlated with hypoxia in all 18 analyzed tumor entities, among them lung, breast, colon and pancreatic carcinoma [13,30]. In this context, numerous studies rank miRNA-210 among the major influenced miRNAs under hypoxic conditions and were able to show a direct regulation of miRNA-210 by a HIF-1α-dependent pathway in PDAC and other cancer entities [35,36]. Diverse impacts on cancer cells could be attributed to miRNA-210 expression, including migration and EMT of PDAC cells [37,38]. Furthermore, a direct binding of 10 consecutive bases in the center portion of miRNA-210 to Regulator of Differentiation 1 (ROD1, also named PTBP3) and a subsequent inhibition of ROD1 was identified [39]. In human PDAC cells a role of ROD1 in gemcitabine resistance is reported as well as an impact of ROD1 expression on cell survival and autophagy levels [40]. In orthotopic mouse models of PDAC, an association of a lack of miRNA-210 and tumor initiation and growth was shown [30].

In recent years, the number of known miRNAs regulated by hypoxia in PDAC has distinctly increased. One of the hypoxia-mediated miRNAs is miRNA-21 which was identified as overexpressed in PDAC as well as in other cancer entities and can be associated with an aggressive tumor disease, poor prognosis, invasiveness and chemoresistance [41]. HIF-1α directly binds to and regulates the expression of miRNA-21. Under normoxic conditions, when HIF-1α is not functional, a miRNA-21 overexpression has nearly no impact on cell proliferation. On the contrary, under hypoxic conditions, when HIF-1α is activated and translocated into the nucleus, an enhanced miRNA-21 expression has a promoting effect on pancreatic cancer cell proliferation [42]. miRNA-21 has also an impact on the apoptotic machinery of cancer cells by upregulating anti-apoptotic B cell lymphoma 2 (Bcl-2) expression and mediating chemoresistance [43]. Due to the fact that miRNA-21 is deregulated in a myriad of cancer entities and due to its impact on cancer development and migration miRNA-21 was classified as an oncogenic miRNA (oncomir) [44].

While the expression levels of the above-mentioned miRNAs were upregulated by hypoxia, there are also studies that report of miRNA-downregulation due to hypoxia. For example, the levels of miRNA-519 showed converse effects: in PDAC cells reduced expression levels of miRNA-519 were shown at low oxygen concentrations. Furthermore, the authors were able to demonstrate a miRNA-519-mediated enhanced invasiveness and tumor growth in a subcutaneous xenograft tumor experiment as well as an interaction with programmed death-ligand 1 (PD-L1) in PDAC cell lines Panc1 and SW1990, which regulated the PD-L1 expression and induced cancer cell apoptosis [45].

miRNA-142 is also down-regulated in PDAC cell lines as well as in PDAC tissues and its expression is inversely correlated with the expression of HIF-1α. In further experiments, a binding of miRNA-142 to the 3′UTR of HIF-1α and a participation of the miRNA-142/HIF-1α axis in the invasion and proliferation of PDAC cells under hypoxic conditions was shown. The analysis of 42 cancer tissues revealed a correlation of HIF-1α and miRNA-142 expression with disease stage and a correlation of miRNA-142 expression with lymphatic metastasis [46]. A summary of the non-coding RNAs involved in hypoxia-mediated cancer progression is given in Table 1.

Cancer cells in comparison to normal human cells release a huge amount of exosomal vesicles that are part of the intracellular communication and play an important role in cancer progression, drug resistance, metastasis and immune evasion [54]. Patton et al. showed that the release of extracellular vesicles by PDAC cells is hypoxia dependent and mediates a survival benefit for cancer cells [55]. Part of the cargo of tumor cell-derived exosomes are miRNAs, so called “exo-miRNAs”, which interact with the surrounding microenvironment, thus mediate an impact on tumor initiation and progression [56]. For instance, hypoxic pancreatic cancer cells secrete miRNA-301a, inducing a M2-polarization of macrophages, thereby promoting malignancy of the cancer cells [57]. Next to the impact of exo-miRNAs on surrounding cells, the exo-miRNAs in biological fluids (serum/plasma) reflect the miRNA signature of the releasing cancer cell. The easy accessibility of these biological fluids enables an analysis of miRNAs in the context as clinically relevant biomarkers [56]. Different studies identified various combinations of miRNA plasma levels to discriminate between cancer patients and cancer-free individuals. Fractions of these differentially expressed miRNAs (e.g., miRNA-155 and miRNA-196, miRNA-181a, miRNA-181b, miRNA-196a, miRNA-20a, miRNA-24, miRNA-25, miRNA-99a, miRNA-21, miRNA-210, miRNA-185, miRNA-191) were hypoxia-regulated (miRNA-21, miRNA-210, miRNA-185, miRNA-191) [41,58,59,60]. In order to improve the accuracy of this diagnostic tool, also in regard to the hypoxia regulated miRNAs, further research has to be done.

### 4.2. Long Coding RNAs and Hypoxia in PDAC

Next to the microRNAs, another group of non-coding RNAs gained attention in normal and pathological cellular functions: long non-coding RNAs (lncRNAs). While miRNAs are defined as non-coding RNAs with approximately 22 nucleotide length, lncRNAs are RNA molecules >200 nucleotides in length [61]. The subcellular localization of the lncRNAs is closely associated with their regulatory mechanisms on several biological processes. Having a nuclear localization, scaffolding and recruitment of proteins and a subsequent formation of 3D nuclear structures influences gene expression. Furthermore, lncRNAs can interact with RNA and DNA molecules and interfere with various parts of mRNA translation and mRNA processing [62]. Having a cytoplasmic localization several lncRNAs can sequester miRNAs and inhibit the binding of miRNAs to their target mRNAs, leading to an upregulation of the target proteins. These interactions have been investigated in a broad spectrum of cancer types including PDAC [63]. NORAD (annotated as LINC00657 in RefSeq) is a lncRNA acting as “RNA sponge” and sequesters miRNAs. Its expression is upregulated during hypoxia in PDAC and this lncRNA promoted EMT by sequestering miRNA-125a-3p. NORAD expression is associated with poor prognosis and metastasis formation in PDAC [53]. In a similar fashion, lowered oxygenation induced the binding of HIF-1α to the HRE of the long non-coding RNA NUTF2P3-001 and upregulated its expression. Upregulated lncRNA-NUTF2P3-001 competitively bound to miRNA-3923 and enhanced the KRAS (Kirsten rat sarcoma 2 viral oncogene homolog) expression which was accompanied by cell survival and proliferation [50]. Under hypoxic conditions the oncogenic lncRNA-FEZF1-AS1 promoted PDAC cell proliferation and invasion through a miRNA-142/HIF-1α axis, while under normoxic conditions a miRNA-133a/EGFR (epidermal growth factor receptor) axis was induced [52].

lncRNAs can also have a direct impact on epigenetic cancer regulation by scaffolding chromatin modifying proteins (e.g., methyltransferases, demethylases, acetyltransferases, and deacetylases) and these complexes can regulate the transcription of nearby (cis-regulation) or genomically distant (trans-regulation) target genes [62,64]. In this context, under hypoxic conditions histone deacetylases are involved in the downregulation of lncRNAs. Liu et al. showed in PDAC cells, that lncRNA-CF129 were transcriptionally down-regulated by the binding of a HIF-1α/HDAC1 complex to its promoter in a hypoxic context. In PDAC tumors a lack of lncRNA-CF129 correlated larger tumor size, lymphatic invasion and metastasis, poor differentiation as well as a decreased overall survival of the patients [22,51].

Another lncRNA which is upregulated in PDAC cells by binding of a hypoxia-activated HIF-1α to a HRE in its promoter is lncRNA-BX111887. This lncRNA promoted the transcription of zinc finger E-box-binding homeobox 1 (ZEB1) and its EMT-associated downstream targets by recruiting the transcription factor Y-box protein (YB1) to the ZEB1 promoter and elevated expressions of lncRNA-BX111887 were associated with tumor progression, late TNM stage, metastasis and lymphatic invasion in PDAC patients [49].

### 4.3. Hypoxia and Its Impact on DNA Modifying Enzymes

Accumulating evidence support the idea that not only HIF-mediated transcriptional changes take place due to hypoxia, but also chromatin remodeling events like DNA methylation and histone modifications are directly affected by hypoxia. In this context, the requirement of a cooperation of epigenetic events with hypoxia-induced transcription factors for a complete initiation of hypoxic pathways or the maintenance of a hypoxic phenotype is taken into consideration [27]. It is generally accepted, that there is a global increase in DNA methylation and histone modifications after a period of hypoxia and, at least in part, this changes can be attributed to HIF-mediated expression of histone modifying enzymes [27,65]. Due to hypoxia, Johnson et al. showed global histone modifications, among them an increased histone3 lysine4 trimethylation (H3K4me3), which is usually associated with enhanced gene transcription or decreased level of histone3 lysine27 trimethylation (H3K27me3), which convey a transcriptional repression [66]. Batie et al. were able to connect the enhancement of hypoxia-mediated H3K4me3 peaks with a EMT signature in HeLa cells [67].

#### 4.3.1. Histone Demethylase

Histone modifications are dynamically regulated by histone demethylases and histone methyltransferases, whereby the activity of some lysine-specific demethylases (KDMs) is oxygen dependent. In this regard, a distinction between KDMs, which are HIF target genes and KDMs, which act as oxygen sensors independent of HIF activity (KDM5A/JARID1A and KDM6a/UTX) has to be made [67,68].

The first group includes the KDM enzymes KDM5B/PLU-1/JARID1B, KDM3A/JHDM2A/JMJD1A, KDM4B/JMJD2B, and KDM4C/JMJD2C whose enhanced expression can be attributed to a direct binding of the transcription factor HIF to a HRE present in their promoter region under hypoxic conditions [69]. Accordingly, the expression of KDM3A/JMJD1A was linked to the expression of HIF-1α in hypoxia in PDAC cells and the results of the study suggested an oncogenic, tumorigenesis promoting function of KDM3A [69]. 

The second group of KMD enzymes acts, as already mentioned, as direct sensors of oxygen. This group includes KDM6A (also known as UTX, ubiquitously transcribed X chromosome tetratricopeptide repeat protein), which acts as a demethylase and forms together with UTY (encoded by the Y chromosome) and KDM6B (encoded by an autosomal gene) the KDM6 enzyme family [68,70]. KDM6A belongs next to the lysine methyltransferases 2C and 2D to the most frequently named mutated epigenetic regulators in cancer, including pancreatic cancer [70]. A loss of expression of KDM6A/UTX was associated with a squamous-like and metastatic subtype of PDAC, especially in females. In a PDAC mouse model, mice with a functional Kdm6a expression showed a suppressed development of tumors while mutant Kdm6a bearing mice developed aggressive, poorly differentiated tumors. Transcriptome analysis form pancreatic cell lines from Kdm6a^-/-^ or wild-type mice revealed an activation of diverse signaling pathways, among them, pathways associated with EMT, proliferation, inflammation and hypoxia [70].

#### 4.3.2. Histone Deacetylases

Next to the histone methyltransferases, HDAC can effectively modulate the accessibility of DNA-coding regions to transcription factors. In conjunction with histone acetylases (HATs), HADC control the lysine acetylation of histone proteins and regulate gene expression by these post-translational histone modifications. HDAC can also act as post-translational modifiers of numerous non-histone proteins, e.g., p53. This non-histone protein modifying activity of HDAC may also be an explanation for numerous side effects HDAC inhibitors (HDACi) exhibited in clinical trials [71]. A suppressed expression of HIF-1α by HDACi under hypoxic conditions suggested that HIF belongs to the group of non-histone proteins under the control of HDAC activity [72,73]. The exact mechanism how HDACi act on HIF-α activity is still not understood, but there are at least 4 possible mechanisms discussed in literature: (1) an inhibition of the nuclear translocation of HIF-α, (2) a destabilization of HIF-α, (3) a repressed DNA binding activity of HIF and (4) a HDACi-mediated repression of the transactivation domain of HIF [74]. In this context, an HDACi-mediated pVHL independent mechanism of proteasomal degradation of HIF-1α was described [75]. The same group also showed that the binding of HIF to its transcriptional coactivator p300/CBP required a deacetylation reaction and p300/CBP seems to be target of this reaction [76]. Furthermore, HIF, p300, HDAC4 and HDAC5 were reported to form a multiprotein complex and the expression of HIF target genes was promoted by this complex formation [74].

In PDAC the expression of HDAC1 was correlated with HIF-1α and MTA1 (metastasis-associated protein 1) protein level and a poor outcome for the patients. The authors described in their study that HDAC1/MTA1 are subunits of the nucleosome remodeling deacetylase (NurD) complex, which is a HDAC-containing repressor complex of proteins with the ability of chromatin remodeling. For this complex the mediation of a HIF-1α regulation was suggested [77,78].

In another study, a participation of HDAC1 in miRNA regulation under hypoxic conditions was described. Zou et al. were able to show, that lowered oxygenation led to the formation of a HIF-1α/HDAC1 complex that bound to the HRE within the miRNA-548an promoter and transcriptionally inhibited the miRNA-548an expression. The inhibition of miRNA-548an induced an upregulation of vimentin mRNA and protein expression and had a promoting effect on cancer progression [32]. In PDAC tissues, the expression of miRNA-548an is inversely correlated with tumor size, tumor stage, appearance of distant metastasis and a poor prognosis for the patient [32].

#### 4.3.3. Polycomb Protein

Polycomb protein complexes are another essential epigenetic system to alter the chromatin structure via histone modifications. The activity of the polycomb repressive complex 2 (PRC2), with its core components EZH1/2 (enhancer of zeste homologue 1/2), EED (embryonic ectoderm development) and SUZ12 (suppressor of zeste 12 homolog) is associated with a transcriptional repression of target genes, especially the silencing of many tumor suppressor genes. Diverse studies described an impact of hypoxia in general and of HIF-1α in particular on elevated EZH2 expressions. In PDAC cells low oxygen concentrations and the resulting activation of the transcription factor HIF led to an enhanced expression of the EMT associated transcription factor TWIST. TWIST, in turn, increased the EZH2 expression and had a promoting effect on tumorigenesis [79]. Similar results were shown in hepatocellular carcinoma cells, were a HIF-1α induced EZH2 expression was associated with an invasive and metastatic phenotype [80]. For an overview of the interaction of HIF and EZH2 in different cancer entities the review by Papale et al. is recommended [81].

In Figure 1 the described epigenetic changes and their impact on PDAC progression are summarized. 

## 5. Final Considerations

As highlighted by this article, by acting as a hypoxic sensor on one hand and being controlled by hypoxia-mediated regulatory pathways on the other hand the epigenetic machinery has a significant impact on hypoxia-associated cell responses. Although the amount of data dealing with the effect of hypoxia on DNA methylation, histone modification or the regulation of non-coding RNAs in cancer in general and in PDAC in particular is continuously rising, the effects of these factors on cancer initiation and cancer progression are only poorly understood. Comparing the data from PDAC with other tumor tissues suggests that these complex regulatory mechanisms might be tissue specific and also dependent on other factors, pointing out the necessity of further studies in more complex models of PDAC. Nevertheless, this tissue specificity in combination with the knowledge of distinctive regulatory pathways might be a basis for the development of new therapy options and identification of specific biomarkers.

## Figures and Tables

**Figure 1 cells-09-02353-f001:**
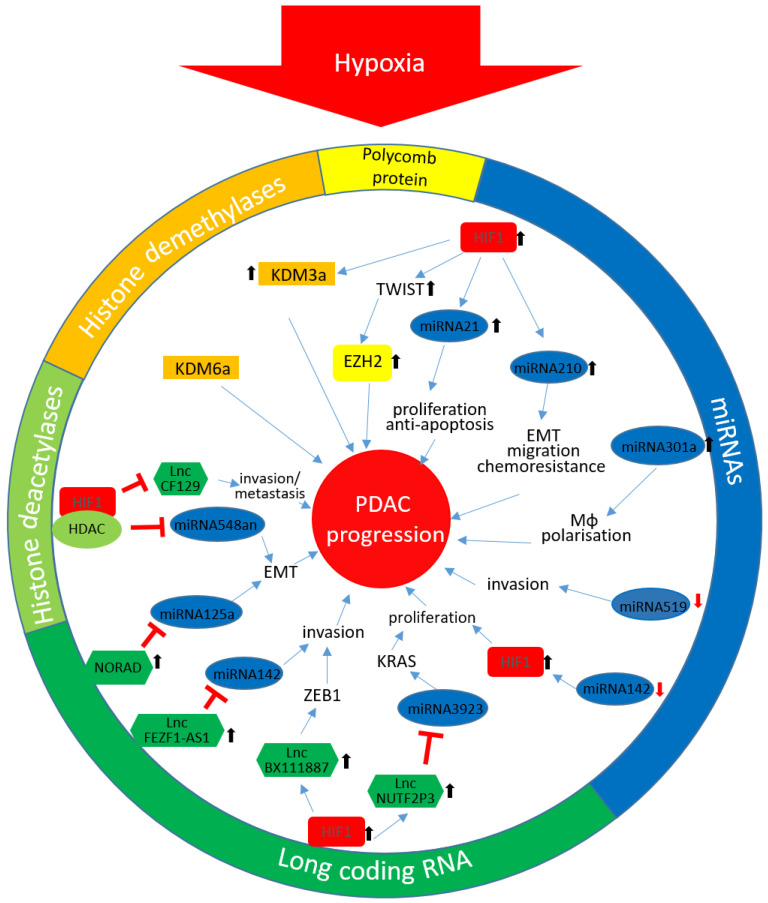
Hypoxia-mediated epigenetic changes in PDAC: The different colours reflect the affiliation to the in the circuit stated coloured epigenetic mechanism. ↑ enhanced expression; ↓ down-regulated expression. HIF-1α, Hypoxia inducible factor 1α; KDM3, lysine-specific demethylase 3; EZH2, enhancer of zeste homolog 2; EMT, epithelial-mesenchymal transition; Mϕ, macrophage; HDAC, histone deacetylase enzyme; KDM6, lysine-specific demethylase 6; lnc, long non-coding; KRAS, Kirsten rat sarcoma 2 viral oncogene homolog; ZEB1, zinc finger E-box-binding homeobox 1; PDAC, pancreatic ductal adenocarcinoma; miRNA, microRNA.

**Table 1 cells-09-02353-t001:** Non-coding RNAs affected by hypoxia in PDAC.

miRNA	Target	Affected Mechanism	Reference
miRNA-21 ↑		proliferation, apoptosis, cell survival	[42]
miRNA-210 ↑	HOXA9	EMT, migration, invasiveness, NF-κB signalling	[38]
mRNA-101 ↓	EZH2	Invasion and metastasis	[47]
miRNA-548an ↓	Vimentin	EMT, proliferation, Invasion	[32]
miRNA-519 ↓	PD-L1	Invasiveness, apoptosis, tumorigenesis	[45]
miRNA-646 ↑	MIIP	Proliferation, invasion, HIF-1a degradation	[48]
miRNA-142 ↓	HIF-1α	Proliferation, invasion	[46]
lncRNA-BX111887 ↑	ZEB1	Proliferation, invasion, migration, EMT	[49]
lncRNA NUTF2P3-001 ↑	miRNA-3923	Cell viability, proliferation, invasion	[50]
lncRNA-CF129 ↓	FOXC2	Cancer progression	[51]
lncRNA-FEZF1-AS1 ↑	miRNA-142	Proliferation	[52]
lncRNA-NORAD ↑	miRNA-125a3p	EMT	[53]

↑ promoted; ↓ inhibited; HOXA9, Homeobox protein Hox-A9; NF-κB, Nuclear Factor kappa-light-chain-enhancer of activated B cells; EZH2, enhancer of zeste homolog 2; EMT, epithelial-mesenchymal transition; PD-L1, programmed death-ligand 1; MIIP, migration and invasion inhibitory protein; HIF1α, Hypoxia-inducible factor 1α; ZEB1, zinc finger E-box-binding homeobox 1; FOXC2, forkhead box C2.

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
