# Peer review of "Coming in the Air: Hypoxia Meets Epigenetics in Pancreatic Cancer"

_cells, 2020, doi:10.3390/cells9112353_

Round 1

Reviewer 1 Report

1. Why there are two affiliations if adress is the same?

"1 Department of Internal Medicine I, Laboratory of Molecular Gastroenterology & Hepatology, UKSH Campus Kiel, Kiel, Germany;

2 Department of Internal Medicine I, Laboratory of Molecular Gastroenterology & Hepatology, UKSH Campus Kiel, Kiel, Germany;"

2. PDAC abbreviation, used throughout the manuscript, comes from pancreatic ductal adenocarcinoma, not only pancreatic adenocarcinoma.

3. What does it mean - "western countries"? The Western world refers to various regions, nations and states, depending on the context.

4. Authors wrote a review without citing any of their work. Do they have experimental publications on this topic? If yes then please make a reference, if not was it entitled to write this review by them?

5. in sentence: "In recent years, the number of miRNAs regulated by hypoxia in PDAC has distinctly increased." there is little misunderstanding - the number is rather the same. It is probably about the number of known mi RNAs sequences

6. "Long conding" should be "Long coding"; "Polycomb proteine" should be "Polycomb protein"

7. Information on the methodology of searching for publications should be added

8. Figure/diagram summarizing the work should be added

9. Do the authors know the publication: "Regulation Is in the Air: The Relationship between Hypoxia and Epigenetics in Cancer" which appeared in Cells? It has some similarity - It should be cited?

Author Response

Reviewer #1:

  1. Why there are two affiliations if address is the same?

Response by the authors:

We are very sorry for the mistake in the affiliations. A.A. has currently two addresses. We corrected the mistake accordingly.

  1. PDAC abbreviation, used throughout the manuscript, comes from pancreatic ductal adenocarcinoma, not only pancreatic adenocarcinoma.

Response by the authors:

We are sorry for the wrong explanation of the abbreviation PDAC and have modified the explanation in the manuscript.

  1. What does it mean - "western countries"? The Western world refers to various regions, nations and states, depending on the context.

Response by the authors:

It is correct to point out that “western countries” is not an unambiguous designation, even though this term is used in various publications for the description of the majority of Europe, Australia and America. Currently, PDAC is the fourth highest cause of cancer-related death in developed countries and predictions estimate negative trends for PDAC-related mortality rates.

  1. Authors wrote a review without citing any of their work. Do they have experimental publications on this topic? If yes then please make a reference, if not was it entitled to write this review by them?

Response by the authors:

The question why an author writes a review about a topic which is until now not part of the authors research interest is obvious. We have a strong background in signalling pathways in PDAC. Over the last couple of years NF-kB, Nrf2 and IER3 were the main topics of our research. Dealing with these proteins, hypoxia and its impact on epigenetic mechanisms in PDAC cells have gained more and more attention to the authors. Especially, the impact of polycomb proteins on EMT associated cell migration under hypoxic conditions is actually one of the research interests of the authors and was starting point to this review article. We hope that this promising project will develop positively and a give rise to an experimental publication in the next couple of months.

  1. in sentence: "In recent years, the number of miRNAs regulated by hypoxia in PDAC has distinctly increased." there is little misunderstanding - the number is rather the same. It is probably about the number of known mi RNAs sequences

Response by the authors:

We are sorry for the unambiguous designation and have changed the manuscript accordingly to the suggestions of the reviewer.

  1. "Long conding" should be "Long coding"; "Polycomb proteine" should be "Polycomb protein"

Response by the authors:

We are sorry for these misspellings and have corrected them.

  1. Information on the methodology of searching for publications should be added

Response by the authors:

This research article has the purpose to give an overview of the literature in the field of hypoxia and epigenetics by collecting data from publications listed in PubMed until September 2020. We included this explanation in the introduction section of the manuscript.

  1. Figure/diagram summarizing the work should be added

Response by the authors:

The authors tried to picture the epigenetic changes in PDAC cells in a graphical abstract.   

  1. Do the authors know the publication: "Regulation Is in the Air: The Relationship between Hypoxia and Epigenetics in Cancer" which appeared in Cells? It has some similarity - It should be cited?

Response by the authors:

Admittedly, the title of the review and the publication “Regulation Is in the Air: The Relationship between Hypoxia and Epigenetics in Cancer” have certain similarities. While the paper by Camuzi et al. deals with the relationship between hypoxia and epigenetic mechanisms and its impact on the acquisition of cancer hallmarks in a variety of cancer entities, the authors of this review focused only on one type of cancer: pancreatic ductal adenocarcinoma. In accordance with the suggestions of the reviewer and due to the content-based similarities, the citation of this excellent and comprehensive review is a useful addition (citation number 28).

Reviewer 2 Report

The review of Claudia Geismann and Alexander Arlt aims to elucidate the intricate relationship between hypoxia and epigenetic mechanisms, and their impact on pancreatic cancer (PDAC).

Hypoxia affects almost all hallmarks of cancer and contributes to a multitude of cellular functions mediating therapy resistance, aggressiveness and metastasis of tumor cells. Increasing evidence suggests that the hypoxic response is codetermined by epigenetic changes in solid tumors, setting the focus of this review on epigenetic changes in PDAC due to hypoxic conditions.

The Authors first focus on illustrating  hypoxia-inducible factors (HIF) a family of transcription factors involved in coordinating the expression of many genes that allow adaptation of the tumor cell to this hostile environment.

Then the Authors discuss how hypoxia influence epigenetic regulation of pancreatic cancer. Epigenetic changes in PDAC in general and under hypoxic conditions in particular are an emerging field of research. These epigenetic changes are characterized on RNA basis, by the regulation of non-coding RNAs (miRNAs and long non-coding RNAs), on DNA basis, by (hydroxy)methylation of DNA as well as on protein basis, by posttranslational modifications of histones or the (in)activation of epigenetic regulator-proteins.

Data presented in this review suggest hypoxia-epigenetics interaction might be tissue-specific and dependent on other factors, such as oxygen deprivation duration. Although such a highly complex interaction is observed during carcinogenesis and tumor progression, modulation of specific players such as histone modifiers presents promising results in cancer cell control in vitro. This highlights the potential of alterations of this regulatory axis to be used as biomarkers of tumor aggressiveness as well as therapeutic targets

The review include a balanced, comprehensive and critical view of the research area. It is well written and easy to read.

Minor points to be addressed:

  1. There are some typos errors as in line 195, 231, 297 etc…..Please, correct
  2. References- ref. 52 and ref.69: page number is missed
  3. Insert more figures could benefit the review.

Author Response

Reviewer #2:

  1. There are some typos errors as in line 195, 231, 297 etc…..Please, correct

Response by the authors:

We are sorry for these misspellings and have checked and corrected the manuscript carefully.

  1. References- ref. 52 and ref.69: page number is missed

Response by the authors:

Since 2016 the MDPI journals use article numbers instead of page numbers. For a unique assignment of the references we supplemented article numbers and not the page numbers to the references.

  1. Insert more figures could benefit the review.

Response by the authors:

The authors summarized the content of the review in a graphical abstract and tried to picture the complex hypoxia mediated regulation pathways in an additional figure.

Reviewer 3 Report

This manuscript by Geismann and Arlt presents a comprehensive review on hypoxia-mediated regulation of epigenetic alterations in pancreatic cancer.

The manuscript is well written, but many sentences are confusing. A revision is needed for improving the sentence structure for better effective communication of the message (proper used of commas and hyphenated compound words). Also, all abbreviations need to be described first, and then be applied throughout the rest manuscript. Most importantly, the authors need to provide a figure schematically summarizing the molecular pathways leading to HIF activation and another for hypoxia-mediated regulatory pathways leading to epigenetic changes during pancreatic cancer progression.

Other revisions requested:

Line 28: “diseases”

Line 84: “Likewise, the effects of radiotherapeutic treatment are highly affected under hypoxic conditions by a reduced DNA free radical availability and an increase in DNA repair enzyme.” Please provide a reference.

Lines 88-89: the sentence is incomplete.

Line 163: Please specify the mouse model.

Line 188: Analyzation is not a word

Line 222: Please replace “cells” by “tumors”.

Line 231: Replace “chances” by “changes”.

Line 243: “Histone demethylase”.

Line 283: “HDAC4 and HDAC5”.

Author Response

Reviewer #3:

  1. The manuscript is well written, but many sentences are confusing. A revision is needed for improving the sentence structurefor better effective communication of the message (proper used of commas and hyphenated compound words).

Response by the authors:

We are very sorry for some complex sentence structure. We carefully revised the manuscript with emphasis on this very important suggestion and proper use of commas.

  1. Also, all abbreviations need to be described first, and then be applied throughout the rest manuscript.

Response by the authors:

We are very sorry and carefully corrected the manuscript accordingly.

  1. Most importantly, the authors need to provide a figure schematically summarizing the molecular pathways leading to HIF activation and another for hypoxia-mediated regulatory pathways leading to epigenetic changes during pancreatic cancer progression.

Response by the authors:

The authors summarized the content of the review in a graphical abstract and tried to picture the complex hypoxia mediated regulation pathways in an additional figure.

Round 2

Reviewer 1 Report

The authors responded to the comments from the review and made appropriate changes. Manuscript may be published in this form